# On Sources of Damping in Water-Hammer

**Alan E. Vardy**

University of Dundee and Dundee Tunnel Research, Dundee DD1 4HN, UK; a.e.vardy@dundee.ac.uk

**Abstract:** Various potential causes of damping of pressure waves in water-hammer-like flows are discussed, with special attention being paid to their qualitative influences on measured pressure histories. A particular purpose is to highlight complications encountered when attempting to interpret causes of unexpected behaviour in pipe systems. For clarity, each potential cause of damping is considered in isolation even though two or more could exist simultaneously in real systems and could even interact. The main phenomena considered herein are skin friction, visco-elasticity, bubbly flows and porous pipe linings. All of these cause dispersive behaviour that can lead to continual reductions in pressure amplitudes. However, not all are dissipative and, in such cases, the possibility of pressure amplification also exists. A similar issue is discussed in the context of fluid–structure interactions. Consideration is also given to wavefront superpositions that can have a strong influence on pressure histories, especially in relatively short pipes that are commonly necessary in laboratory experiments. For completeness, attention is drawn towards numerical damping in simulations and to a physical phenomenon that has previously been wrongly cited as a cause of significant damping.

**Keywords:** damping; attenuation; water-hammer; pressure waves; wavespeed; unsteady friction; fluid–structure interaction; wave superposition; valve closure

## 1. Introduction

The root causes of pressure surges in pipeline systems are well understood and the analysis of them is a relatively mature subject. At first sight, it might be expected that this would imply that systems can always be designed with high confidence whereas, in practice, this is not the case. Likewise, it might be expected that, if a system does not behave in the originally intended manner, it should be relatively straightforward to infer the reason why. Again, however, this is not always the case; indeed, it is rarely so. One reason for this is that many different phenomena can occur simultaneously and interact with one another. Even if an engineer is well familiar with each individual phenomenon, it may be difficult to predict the consequences of interactions or to recognise their relative importance—or even their existence—in systems that are behaving in an unexpected manner.

In both cases, the task can be especially challenging when damping phenomena cause significant changes in the amplitudes and/or shapes of the wavefronts propagating through the systems. Accordingly, it is beneficial to have a clear understanding of the potential sources of damping and their likely importance in any particular situation. This paper is an attempt to summarise the common causes and to give a degree of guidance on when they are likely to be significant. It is not intended as a review, and neither is it intended to be comprehensive. These objectives would not be practicable, partly because of space limitations, but also because of the author's own ignorance, which has frequently been exposed when attempting to unravel secrets that nature has hidden in unforeseen behaviour. Indeed, identifying the causes of unwanted behaviour can be highly challenging even for persons with much wider experience than the author.

### 1.1. Damping

The focus of this paper is on the so-called damping of pressure waves propagating in an essentially one-dimensional (1-D) manner. For simplicity, most of the descriptions

refer to pipes and these are assumed to be of a circular cross-section. However, many of the phenomena are also observed in other contexts, notably free-surface flows. The term 'damping' is used generically. It usually implies a continuing reduction in amplitudes as time evolves and it often also implies changes in the shapes of waves as they propagate. Commonly, it is a consequence of phenomena that cause significant dissipation, but this is not always the case. The examples given below also include cases where it results from dispersive behaviour that, at least as a first approximation, simply redistributes energy. This is not a trivial distinction. Dissipative phenomena always cause damping, and dispersive phenomena often do so, but there are exceptions in which re-distributions of energy cause superpositions that would not otherwise have occurred. Examples of this are given below.

In the context of water-hammer in pipelines, damping is usually beneficial. It can enable strong disturbances to decay before the arrival of subsequent disturbances, thereby reducing the potential consequences of superpositions. However, it is not always helpful. For example, it complicates the application of leak-detection techniques based on intentionally created pressure waves. Even when the damping is not strong enough to prevent good estimates of the locations of leaks or restrictions, etc., it may downgrade the reliability of estimations of their sizes.

### 1.2. Outline of Paper

Most of this paper is devoted to presenting the causes of damping and to summarising their consequences. Thereafter, attention is focussed on the use of this knowledge in the assessment of measured pressures. In such cases, the outcomes are known, and the challenge is to deduce what caused them. To avoid unnecessary complications, the discussion centres on the flows in two simple geometrical configurations. One is the almost trivial case of a single pipe along which a wave is travelling. The other is the classical reservoir-pipe-valve system typically chosen in many studies of water-hammer. The first of these is adequate for studies of phenomena in which significant damping occurs during the propagation of individual waves. The second is more representative of studies undertaken in laboratories. Typical practical applications include additional complications, of course, such as networks, pumps, air valves, etc., but their inclusion herein would complicate and extend this paper unnecessarily. Likewise, attention is focussed exclusively on individual causes of damping, not on the many possible combinations thereof. This is important for clarity of presentation, and it is also realistic. Although it will be common for two or more causes of damping to exist simultaneously, it will be usual for one of these to be dominant and the others to be of secondary importance.

For completeness, it is declared that the remainder of this paper focusses exclusively on waves that may be regarded as travelling in a plane-wave manner. That is, they can be represented with sufficient accuracy without taking account of the variations in pressure over the pipe cross-section (even though lateral variations in axial velocity will exist). This limitation will rarely, if ever, have significant consequences for persons assessing waves caused by strong flow disturbances of the type that are usual in water-hammer-like flows, i.e., resulting from events such as valve closure or pump trip. In such cases, low-frequency components of waves are dominant. In some other types of application, however, high-frequency components can be dominant. This is commonly the case in studies of acoustics, and it is also important in specialist applications such as leak detection in pipelines. When waves are of a sufficiently high frequency for their wavelengths to be shorter than a few pipe diameters, cross-sectional variations in pressure become significant. Indeed, in the case of waves with frequencies greater than well-known cut-off frequencies, radial disturbances dominate axial ones. Many possible modes of vibration exist simultaneously at any frequency, but plane-wave behaviour is strongly dominant at frequencies smaller than, say, a quarter of the lowest cut-off frequency. Thereafter, radial modes rapidly increase in importance and plane-wave components decay rapidly.

The scope of this paper is also limited to interpretations in the time domain. Again, this is appropriate for the water-hammer-like flows at which this paper is targeted and

in which low frequencies are dominant. In principle, however, it is possible to convert time-domain histories into frequency-domain spectra and to interpret everything in the context of frequency-dependence. This can be advantageous in particular applications, such as leak and blockage detection, even when attention is directed primarily at frequencies that are well below cut-off frequencies. It would be useful for another paper to discuss damping in applications in which frequency-domain interpretations are more instructive than time-domain ones. However, such a paper would need to be written by a different author with a sufficiently deep knowledge of the assessment of frequency-dependent damping in relevant applications.

Throughout this paper, the focus is on readily observable flow behaviour or, more strictly, on readily measurable properties of the flow, especially on pressure histories. These can be measured with a good accuracy by surface-mounted sensors, whereas measurements of parameters such as velocity and shear stress are not normally available in practical engineering applications with rapidly varying flows. Indeed, even if they were, it is unlikely that they would simplify interpretations greatly. This is because strong variations can exist even in individual cross-sections at a single instant in time. For them to add significantly to the usefulness of pressure measurements, it would be necessary to be able to deduce their average values (e.g., cross-sectional flow rates, not just point velocities). For analogous reasons, this paper does not attempt to use energy-based methods to assess damping even though they can be used to good effect in theoretical modelling (e.g., Karney [1], Axworthy et al. [2]).

## 2. Skin Friction

The most well-known cause of damping is skin friction. Its influence can be illustrated in a simple manner by considering an instantaneous step change in pressure at one end of a semi-infinite pipe. In the absence of friction (and all other complications), the wavefront would propagate in an unchanged form, illustrated by the dotted line in Figure 1. If, instead, the existence of skin friction is acknowledged, the pressure will decrease downstream because a longitudinal pressure gradient is needed to counter the resistance at the wall. The greater the distance travelled by the wavefront, the greater the overall resistance and so the pressure at the wavefront reduces continually. This effect was discussed by Leslie and Tijsseling [3] in the context of a wavefront generated by a valve closure. The reducing amplitude of the step at the leading edge of the wave implies a reducing change in the velocity. That is, the flow is decelerating, and the magnitude of the pressure gradient is influenced by this.

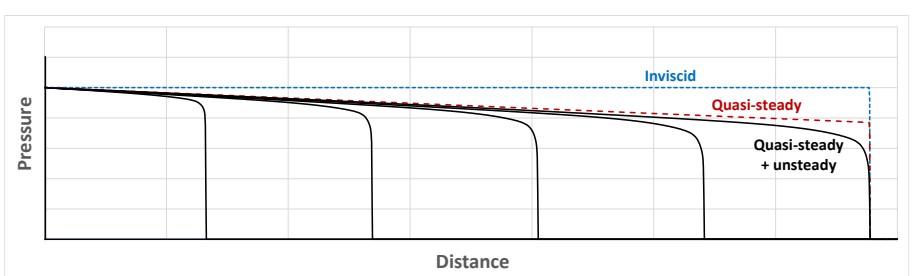

**Figure 1.** Influence of skin friction on a compressive wavefront in pipe flow.

A first approximation to the magnitude of the evolving shear stress at any location can be obtained by assuming that, at any instant, the velocity distribution over the cross-section is identical to that in a steady flow. In this case, the wall shear stress will be deduced from the steady-flow relationships and the resulting pressure distribution will be as indicated in Figure 1 by the broken line labelled 'quasi-steady' friction. Strictly, even for the single wavefront under consideration, the assumption of quasi-steady behaviour does not imply that the velocity behind the wavefront is axially uniform (although it will often be nearly so). The reason for this is that the continual increase in resistance resulting from the increasing

distance travelled by the wavefront causes a continual reduction in the pressure change at the wavefront and, hence, also of the velocity change. Such changes are transmitted back to the upstream boundary at the speed of sound and their reflections at the boundary travel back downstream, albeit at almost the same speed as the wavefront itself. For completeness, it is noted that even the speed of the propagation of the wavefront itself does not remain exactly constant, although the variation is of negligible consequence in almost all liquid flows.

The assumption that skin friction behaves in a quasi-steady manner is, at best, an approximation. In reality, at any location, the response of the velocity profile to a change in the flow rate is time-dependent. The outermost regions of the cross-section respond first and vorticity diffusion gradually causes consequential changes in the core region of the flow. The times required for the radial vorticity diffusion to cause the necessary adjustments of the velocity profiles are far greater than those required for the wavefronts to propagate over similar distances. This leads to pressure distributions resembling the continuous lines in Figure 1. The greater the distance of any particular location behind the step, the greater the elapsed time since the step passed that location. Therefore, the greater the time during which the velocity profile has been adjusting, the closer its approximation to a quasi-steady form. The processes of vorticity diffusion are somewhat different in laminar and turbulent flows and the consequences of this can be significant. Ghidaoui and Kolyshkin [4] discuss this in detail and show that it is responsible for differences between the small-time effects seen in Figure 2 below and longer-term effects measured in some other experiments, e.g., Shuy [5].

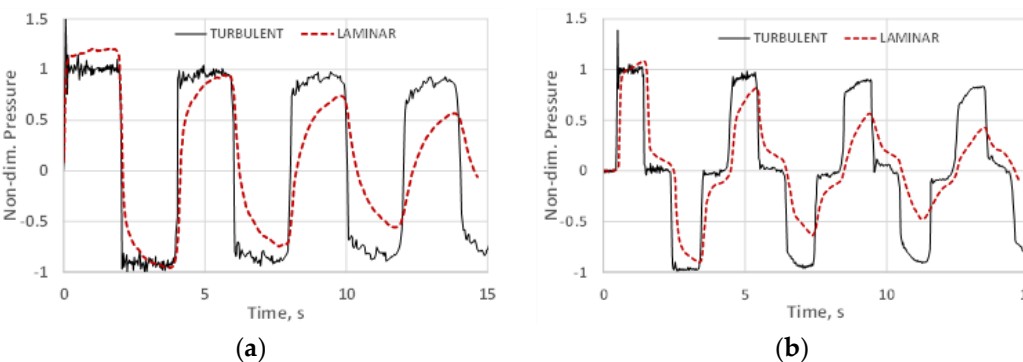

**Figure 2.** Measured pressures following rapid valve closure in a laboratory pipe (after Holmboe, [6]). (**a**) At the valve. (**b**) At mid-pipe.

The strength of this unsteady effect at any instant depends upon the rates of change in the flow rate, not only at the current instant, but also before this instant because of the time needed for vorticity diffusion. These are strong during the passage of a wavefront and much weaker after its passage and this is the root cause of the characteristic shape of the wavefront seen in Figure 1. This shape is commonly seen in measurements made in laboratory studies of the phenomenon, but it is usually less pronounced in full-scale measurements. There is a simple reason for this. Although wavefronts generated by events such as a valve closure or pump-start-up can be quite compact, no such generalisation can be made about the time intervals between them because these depend on the overall system, not on local phenomena. As pointed out by Duan et al. [7], pipe length:diameter ($L/D$) ratios are important in this respect. These tend to be much longer in full-scale engineering applications than in laboratory experiments, so the time intervals between the successive reflections of any particular wavefront are correspondingly larger. Overall, therefore, the relative importance of unsteady-friction effects is usually much greater in laboratory investigations than in most practical applications. It is somewhat ironic that the need for researchers to understand the consequences of unsteady friction exists, in part, because their experiments cannot reproduce full-scale $L/D$ ratios.

*Measured Pressures*

Quasi-steady friction in pipe flows is well understood and widely available representations of it are adequate for most engineering purposes. The position with unsteady friction is less secure, but it is not the purpose of this paper to review analytical models of it. Instead, the key is to be able to recognise it in physical measurements. In practice, this has to be done at second-hand through measurements of pressures because shear stresses are almost never measured directly except in highly specialised laboratory experiments designed expressly for this purpose (e.g., Shuy [5], Vardy et al. [8]).

Figure 2 shows well-known experimental measurements obtained by Holmboe [6] in a reservoir-pipe-valve system. These experiments were reported by Holmboe and Rouleau [9] and used by Zielke [10] to assess his ground-breaking model of unsteady friction in laminar flows. Because the experiments were designed explicitly to assess the influences of skin friction, the pipe was encased in concrete, thereby eliminating any risk of significant influences of pipe movement. The pressure disturbances were caused by sudden closures of the valve following a period of steady flow. This caused steep pressure wavefronts to travel upstream towards the reservoir and the figure shows such a wavefront together with successive wavefronts after reflections at both ends of the pipe. The left-hand box shows pressures just upstream of the valve and the right-hand box shows pressures at the mid-length of the pipe. Both laminar flow and turbulent flow cases are shown and, to facilitate comparisons, the pressures are scaled by the Joukowsky pressure.

Consider first the turbulent flow case. Temporarily disregarding the sharp peak, the initial pressure rise is a close approximation to a step increase and the subsequent reflections after the wavefront has propagated back and forth along the pipe are almost equally steep. During the early periods, the pressure between successive reflections is nearly constant, but it becomes progressively less so with increasing time. The reason for this can be inferred from the corresponding measurements for the laminar flow case. The first reflection clearly exhibits the behaviour described above in relation to unsteady friction and the distortion becomes increasingly pronounced with each subsequent reflection. Indeed, the distortion is so strong that it visibly affects the shape of the whole of the wavefront, even in the first reflection. With this basis for comparison, it is possible to infer with high confidence that the similar, but much less pronounced, trend in the turbulent flow measurements is also attributable to unsteady friction. In both cases, the pressures between successive passes of the leading tips of the wavefronts can be interpreted as approaching quasi-steady values asymptotically.

For completeness, brief attention should be paid to the slightly noisy nature of the turbulent flow measurements. Although other explanations are possible, it seems likely that these are attributable to the pressure sensors, not to the flow itself. This inference is supported by the fact that the noise is consistently greater at the valve than at the mid-pipe. The strong initial overshoot in the turbulent flow traces is also assumed to be due to this cause. The only other possibility envisaged by the present author, namely a very strong axial vibration of the valve, seems implausible.

In both the laminar and turbulent cases, the peak positive and negative values decrease monotonically in time. This is a natural consequence of the dissipative influences of viscous phenomena, both quasi-steady and unsteady. However, it is also seen that the decay in amplitudes is stronger at the mid-pipe than at the valve, especially in the laminar flow case. At first sight, this might seem counter-intuitive, but it is a natural consequence of the ever-increasing length in the wavefront that has been influenced significantly by the unsteady component. When this length exceeds the length of the pipe, the superposition of the wavefronts propagating in opposite directions prevents the development of the complete pressure changes associated with the individual wavefronts. This effect is discussed more fully in Section 8 below.

Holmboe's experiments were close to ideal for researchers studying unsteady friction. However, it is emphasised that the influence of unsteady components of skin friction in practical applications is less strong than that seen in Figure 2. There are three reasons

for this. First, the measurements were made in a pipe with a very small length:diameter ratio so the time intervals between successive steps are short. Second, the valve closure was as rapid as the experimenters could achieve. Third, the Reynolds numbers of the initial steady flows were smaller than will be usual in large-scale engineering. The second and third of these factors are highly desirable for researchers studying unsteady friction and even the first has advantages. Nevertheless, it must not be inferred that the overall influence of unsteady friction in practical pipe systems is so strong. It is indeed strong close to sudden wavefronts, but highly abrupt wavefronts are less common in practice and the time intervals between successive wavefronts are usually much greater than those required for vorticity diffusion over a pipe's cross-section.

Figure 3 shows theoretical predictions of pressure histories from a study described by Vítkovský et al. [11] caused by rapid and slow valve closures in a turbulent pipe flow. This enables the relative influences of the quasi-steady (QS) and unsteady (US) contributions to be assessed. Using the discussion on Figure 1 above, it can be inferred that the damping is caused primarily by the quasi-steady component, whereas the distortion of the leading part of the wavefront is caused primarily by the unsteady component. Moreover, in common with Figure 2 above, successive wavefront reflections arrive at intervals that are shorter than those needed for vorticity diffusion to enable cross-sectional velocity profiles to approach quasi-steady conditions closely. With the slower valve closure, the differences between predictions with and without unsteady friction are less obvious visually than for the rapid closure. This is consistent with expectations discussed above in relation to timescales, but these histories are also strongly influenced by the superpositions of successive wavefronts.

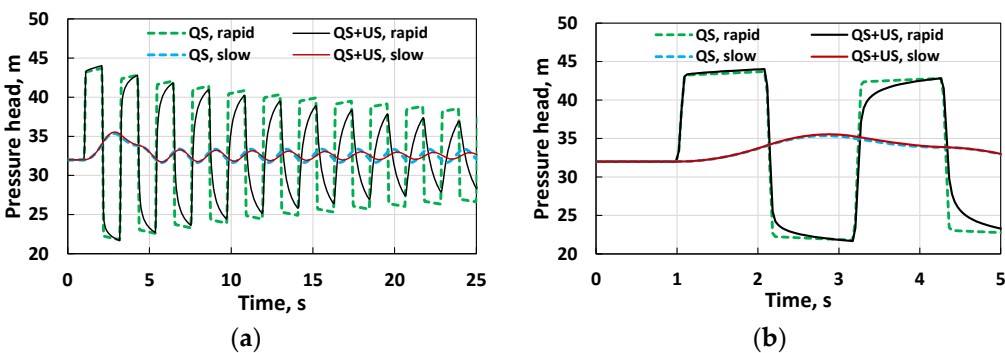

(a)             (b)

**Figure 3.** Influence of skin friction on simulated pressures in an unsteady, turbulent, pipe flow induced by rapid and slow valve closures. (**a**) Progressive decay. (**b**) Detailed form.

## 3. Pipe Wall Properties

In the preceding section, no account was taken of pipe wall flexibility. This is sometimes a close approximation to reality—tunnels, pipes encased in concrete, etc. In such cases, the wavespeed is controlled almost exclusively by the fluid properties. For water at normal pressures and temperatures, this is of the order of 1450 m/s. At the other extreme, e.g., in blood vessels, the wavespeed may be controlled almost exclusively by the vessel wall properties. In this section, attention is focussed on pipes for which the wavespeeds are determined mostly by the fluid properties, but the influence of pipe flexibility is nevertheless significant. Nevertheless, it is assumed that the wall stresses are in radial equilibrium with the fluid pressure at all times. That is, radial inertia is disregarded. This is a good approximation except in the case of very high frequency fluctuations.

The simplest case is characterised by steel pipes that are thin-walled, but of sufficient thickness to withstand all water-hammer pressures that could occur. In this case, when a sudden event such as rapid valve closure occurs, the principal water-hammer wavefront is somewhat smaller than in a rigid pipe because the increased pressure causes a small increase in the cross-sectional area. In effect, the expansion reduces the overall stiffness, thereby reducing the wavespeed and, hence, also the pressure rise needed to cause the required change in the velocity. In most applications, it is sufficient to analyse such flows

in the same manner as for a nominally rigid pipe except for using a reduced, but constant, wavespeed. The expected pressure histories then differ from those in rigid pipes only by reduced amplitudes and by increased intervals between successive reflections.

Although usually unnecessary, a more rigorous assessment of the influence of pipe elasticity would acknowledge that, because of Poisson's Ratio effects, the circumferential stresses that resist radial expansion must give rise to axial stresses. This topic is addressed in Section 4. First, however, consideration is given to the waves in pipes that respond gradually to changes in the fluid pressure.

*Inelastic Pipe Walls*

Steel pipes (and other metal pipes) are conventionally treated as linearly elastic in studies of water-hammer. PVC pipes, however, are far from linearly elastic and yet they are commonplace in practical applications such as water supply. The wall material of such pipes can behave in a visco-elastic manner. That is, the pipe diameter responds relatively slowly to changes in the fluid pressure. Imagine, for instance, a sudden, sustained increase in the fluid pressure. The apparent stiffness associated with the initial pipe response to this is greater than the subsequent apparent stiffness, which falls asymptotically to a value that, in the absence of a further pressure change, is maintained indefinitely. The timescales associated with this relaxation are typically much greater than those associated with vorticity diffusion over a pipe cross-section, so the distortion of the pressure wavefronts is also much greater.

The consequences of this for water-hammer phenomena are illustrated in Figure 4, which again uses data obtained in the study reported by Vítkovský et al. [11]. The graphs showing the influence of quasi-steady (QS) and unsteady (US) components of skin friction are the same as those in Figure 4. By inspection, these are much less important than the influence of the assumed visco-elastic (VE) properties of the pipe wall. Indeed, the latter is so dominant that little useful purpose would be served by taking account of unsteady components of skin friction in analyses of water-hammer in pipes with wall properties such as these (although quasi-steady friction will still be influential in long pipes). In a research context, it will be rare for reliable deductions to be made about unsteady friction from measurements made in such pipes. This is the case even though the prevailing shear stresses will be comparable to those in nominally rigid or linearly elastic pipes. However, it is also true that the existence of unsteady friction will reduce somewhat the accuracy with which the visco-elastic effects themselves can be deduced.

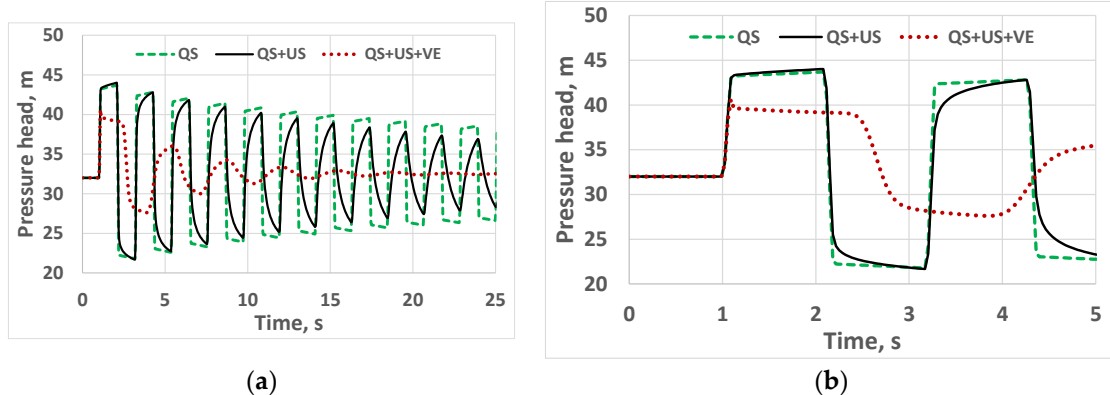

(a)  (b)

**Figure 4.** Influence of visco-elasticity on simulated pressures in an unsteady, turbulent, pipe flow induced by rapid valve closure. (**a**) Progressive decay. (**b**) Detailed form.

In passing, it is informative to consider the implications of figures such as Figure 4 for researchers attempting to calibrate numerical models of the consequences of visco-elasticity from measurements in relatively short pipes used typically in laboratory experiments. As seen in the preceding section, even with the shorter delays caused by unsteady fric-

tion, wavefronts can soon begin to overlap with their own reflections from boundaries. As a consequence, only the early stages of the phenomenon can be calibrated directly. The increased timescales associated with visco-elasticity can be a major complication in this respect.

A related difficulty illustrated in Figure 4 arises when wavespeeds need to be inferred from experimental measurements. In this context, however, the appropriate interpretation of the term 'wavespeed' tends to be context-dependent. As discussed by Tijsseling and Vardy [12], it is important to distinguish between speeds at which small disturbances travel (i.e., the speed of sound relative to the fluid) and the speed at which large disturbances appear to travel (e.g., distances moved by wave crests in a given time). Using the latter, the intervals between successive reflections in the elastic-walled pipe case are clearly distinguishable from the instants when the pressures begin to fall. The same is not true, however, in the case of visco-elastic pipes. The interval between the start of the first two maxima is reasonably distinct, but the intervals between the succeeding maxima are less distinct. Moreover, the intervals between the successive maxima increase with time and the extent of the overlap between each wave and its reflection also increases. As a consequence, if physical measurements were of this form, it would not be safe to infer the wavespeeds from the time intervals between the maxima. When such behaviour is interpreted in the frequency domain instead of the time domain, the wavespeed is said to be frequency-dependent (e.g., Aliabadi et al. [13]

## 4. Axial and Lateral Structural Movement

Attention now turns to the consequences of Poisson's Ratio effects in pipe walls, and linearly elastic behaviour is again assumed. In the event of a sudden disturbance to the flow, e.g., valve closure, the change in the internal pressure causes a change in the circumferential stress in the pipe wall and, hence, also causes a change in the axial stress in the wall. This change propagates axially in the wall at the speed of sound in the solid material. Typically, as illustrated in Figure 5, it travels much faster than the pressure wave in the fluid. In turn, the Poisson's Ratio effects at the axial-stress wavefront cause a change in the circumferential stress that changes the pipe diameter and, hence, induces a pressure change in the fluid. This form of fluid–structure interaction (FSI) is an example of a dispersive process that is not necessarily dissipative (at least, not strongly so).

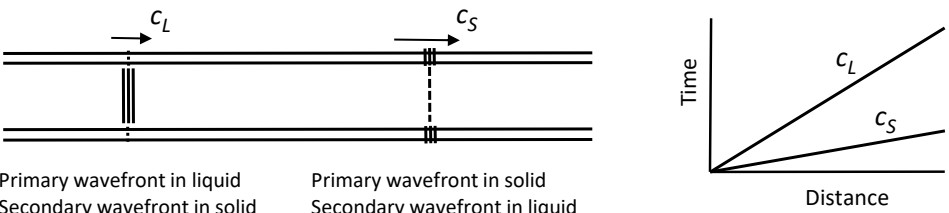

**Figure 5.** Wave pair in fluid–structure interaction.

In the context of water-hammer, the pressure wavefront associated with the axial stress wave is usually known as a precursor wave and the pressure change that accompanies it is typically much smaller than that in the primary wavefront. Usually, it is of secondary importance, but, as illustrated below, this is not always the case. This depends upon how the two wavefronts behave when they meet a pipe boundary. Whenever either type of wavefront reaches an end of a pipe, it causes two reflected wavefronts, namely one of each type. Likewise, at a junction of two or more pipes, it causes two wavefronts to propagate along each pipe connected to the junction. It is possible for this to cause cumulative reductions in the amplitudes of the principal wavefronts that, especially in networks, can have similar consequences to conventional damping. However, it is also possible for the effect to cause greater changes in the pressure in some locations than those in the initial

disturbance. Therefore, it is best to regard it as a fragmentation process that will commonly lead to reduced peak-to-peak amplitudes rather than to regard it as damping per se.

Phenomena associated with changes in pipe cross-sectional areas are far from being the only potential structural influences on pressure histories in pipes. Other possibilities include, for instance, the non-rigid behaviour of supposedly rigid restraints and strong vibrations in suspended pipes. In most applications, the freedom of pipes to move is relatively small and, in this case, the influence of the movements is likely to be significant only relatively close to the wavefronts. Nevertheless, the consequences of even small movements can be difficult to unravel from other, longer-lasting effects. It is, therefore, useful to remember that supports designed to exercise restraint as a consequence of their stiffness do not exert any restraint at all until movement actually occurs. In principle, movements are possible either normal to a pipe axis or parallel to it (or both). Flexural motion is usually easier to restrain than axial motion, but this is not always true. In any case, supports that are capable of effectively preventing significant lateral movements might be much less effective in preventing axial movements, e.g., Tijsseling and Vardy [14]. Both types of movement are readily initiated when the pressure waves meet changes in the axial orientation of a pipe, at bends and T-junctions, for instance. This can have especially strong consequences in suspended pipe systems, but it is also relevant to pipes supported on the ground or even buried in the ground.

So far, it has been implicitly assumed that the initiating event primarily causes a disturbance in the fluid and that the consequential response of the pipe is of secondary importance. However, this is not always the case. Externally induced shaking of pipes, e.g., in earthquakes, causes structural movements that can have strong consequences for the pressure change. Figure 6 shows an extreme example described by Vardy et al. [15] in which the initiating disturbance was induced structurally, not hydraulically, by the axial impact at a closed end of a pipe. The main experiments were undertaken with the T-piece configuration shown in Figure 6a, but preliminary measurements were made with only a single pipe, with the junction replaced by a closed end. Figure 6b shows the pressure measurements at the mid-length of the single pipe. Waves in the liquid and solid are denoted by 'L' and 'S', respectively. By inspection, the initial axial stress wave (S1) arrives long before the pressure wave (L1), and so does the reflection (S2) of the stress wave from the remote end. Its subsequent reflection (S3) from the impact end arrives shortly afterwards. The pressure wavefront L2 is a disturbance moving upstream, having been generated when the stress wave S1 reflected at the remote end. It is noteworthy that the magnitude of this wavefront exceeds that of L1. A further important feature of the figure is that the maximum and minimum pressures occur long after the initial pressure wave and its first reflection. This illustrates the potential for a dispersive effect to cause amplification, not damping.

Figure 6c shows the pressure at the main junction in the full T-piece configurations. The first event, labelled L2*, is the response of the pressure to the reflection that, in the axial-only configuration, gave rise to L2 and S2 on arrival of the first stress wave (S1). Likewise, the event labelled L2** corresponds to the arrival of the stress waves from the remote ends of the branches. This is followed closely by the arrival of the first pressure wavefront (L1) from the initial impact. The gradual reduction in pressure during the intervening period is a consequence of flexural movements in the branches. As this develops, the lateral movement of the branches at the junction corresponds to an elongation of the axial pipe. As stated above, this is an extreme example of the possible consequences of FSI, because it is caused by an external structural force. Nevertheless, it demonstrates that structural movement can cause behaviour that will complicate attempts to make reliable inferences from the measurements of pressure alone.

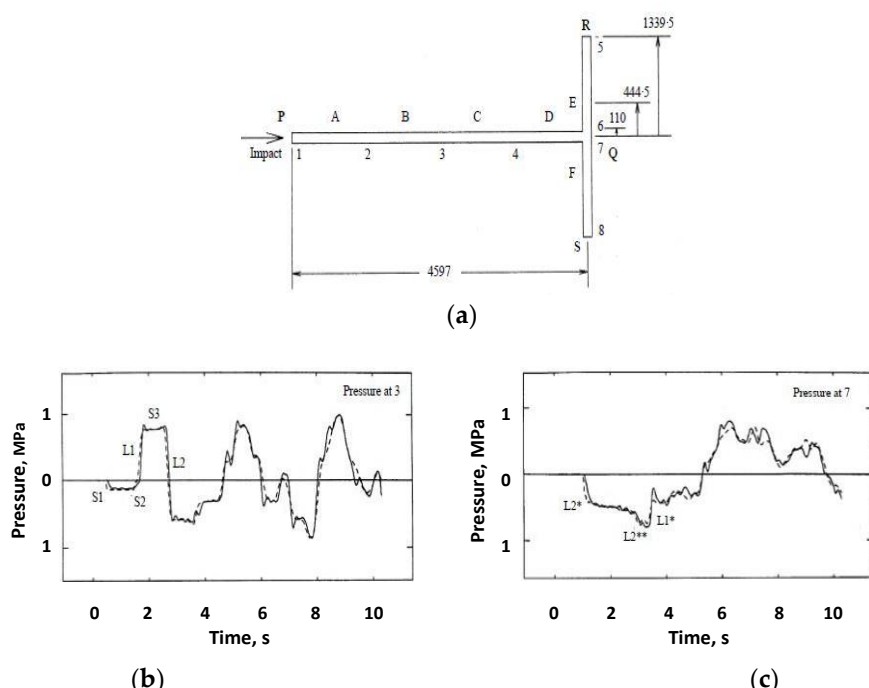

**Figure 6.** Influence of axial and flexural stresses in pipe walls (after Vardy et al. [15]). (**a**) T-piece experiment. (**b**) Axial pipe only. (**c**) Full T-piece configuration.

## 5. Variable Wavespeed

As is well known, the relationship between changes in the pressure and velocity associated with pressure waves is strongly dependent on the speed of sound. An obvious consequence of this is that any phenomenon that can cause changes in the speed of sound will also cause dispersive behaviour and, hence, influence damping (either increasing or countering it). An obvious example arises when free gas bubbles exist in a liquid. Even very small proportions of gas can have a strong influence because they can hugely increase the overall compressibility of the fluid whilst having a minimal effect on its bulk density.

Consider first a system in which the gas is distributed uniformly. In this case, pressure waves causing only small proportional changes in the absolute pressure would have little influence on the wavespeed so this would be almost uniform throughout. The dominant difference from the same system with a wholly liquid flow would be a reduction in the pressure changes needed to cause any particular changes in the flow. However, in neglecting other effects, such as skin friction, there would be neither dissipation nor significant dispersion. A different picture exists when the pressure changes are large enough to cause significant changes in the bubble volume because this changes the local compressibility and, hence, also the local wavespeeds. As a consequence, it is not possible for a strong wavefront to travel along a pipe in an unchanged form.

This effect is illustrated in Figure 7 for an almost equivalent compressible phenomenon, namely pressure wave propagation in an ideal gas. This is a convenient choice, partly because of the author's background, but also because the relationships between the density, sound speed and pressure are especially simple. To highlight the effect under discussion, all other causes of change except quasi-steady friction are suppressed; e.g., the pipe is rigid, unsteady skin friction is neglected and there are no heat transfers between the fluid and the pipe. The figure shows the pressure and velocity histories at regularly spaced locations along a pipe when the pressure at its upstream end is increased suddenly to a new value and is then held constant. The continuous and broken lines show conditions with upstream pressure increases of 1% and 10% of ambient pressure, respectively. In both cases, the velocity reaches a maximum immediately behind the wavefront and then gradually decays. In contrast, the pressure increases continuously after the wavefront passes. This behaviour

exists in both cases, and it is especially pronounced in the case with the larger increase in the pressure. Even in this case, however, the proportional increase in the speed of sound across the wavefront is only about 1.3%. Much greater changes can be expected in the case of liquid flows with free gas bubbles.

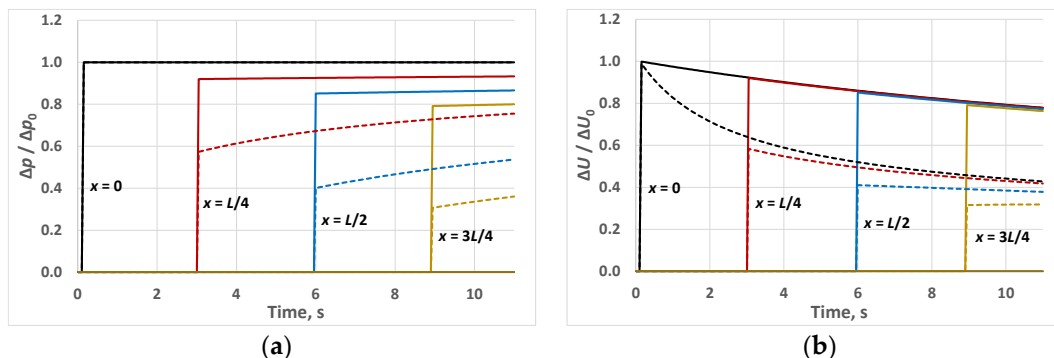

**Figure 7.** Influence of fluid compressibility on wavefront propagation. (Continuous lines: $\Delta p_0 = 1$ kPa; Broken lines: $\Delta p_0 = 10$ kPa.). (**a**) Pressure. (**b**) Velocity.

In the case of bubbly flows, it will rarely be reasonable to expect the gas to be distributed uniformly throughout a system. When this is not the case, spatial variations in the speed of sound will exist even in ambient conditions and these will further complicate the propagation of waves. As an extreme example, consider a local region of uniformly distributed bubbles in an otherwise gas-free system. When a pressure wave arrives at an interface between the different zones, it will partially reflect in a manner that resembles the corresponding behaviour on arrival at a junction between two pipes with different diameters. The equivalent behaviour will exist continuously along regions with gradual variations in the bubble content. The wavefronts propagating into regions of increasing gas will flatten and the wavefronts propagating in the other direction will steepen. In both cases, however, the process will cause continuous reflections that will influence conditions elsewhere in the system. In some locations, this will appear to be damping, whereas, in others, it will have the opposite effect. Such behaviour will inevitably complicate the task of anyone attempting to interpret the measured pressure signals without prior knowledge of the processes giving rise to them.

Even greater complexity exists when spatially varied distributions of bubbles convect through a system because this changes the system response to identical initiating disturbances. All effects, including the strength of the damping behaviour, are affected. However, the possible variations are too great to consider in an overview paper such as this. Likewise, no attention is paid to even more extreme conditions associated with phenomena such as cavitation, column separation or pressurised free-surface flows, all of which can also cause other behaviours that are significantly more important than the damping effects that they might induce.

## 6. Porous Surfaces

Strong damping can exist when fluid can discharge laterally through pipe walls. It will be rare for this to be desirable in the case of liquid flows, but it can be beneficial in some gas flows. Well-known examples are gun silencers and vehicle exhausts. Usually, however, both of these exist over only short lengths of pipe. To add variety, the author takes this opportunity to indulge himself by addressing a topic that has been important in his own career, namely pressure wavefronts generated when trains enter tunnels at high speed. Such wavefronts can be characterised as a relatively steep ramp; e.g., a pressure rise of, say, 2 kPa in less than 10 tunnel diameters followed by an extended, more gentle ramp. The initial steep ramp is generated during the short period when a train nose suddenly

causes a partial blockage of the tunnel portal. The subsequent, more gradual rise develops as the length of train inside the tunnel increases, causing increasing frictional resistance.

In tunnels of slab-track construction, i.e., with relatively smooth and impervious surfaces over the whole cross-sectional perimeter (including the track bed), a wave-steepening effect seen in Figure 8a exists and, in the absence of effective countermeasures, it can lead to the emission of unacceptable sonic boom-like disturbances from tunnel exit portals. Methods of countering this behaviour are outwith the scope of this paper and, instead, attention is focussed on the corresponding outcome in ballast-track tunnels (tracks mounted on deep layers of coarse gravel). In such tunnels, the wavefronts initially steepen, but after a sufficient distance, they begin to flatten, as seen in Figure 8b. The mechanism causing the strong influence of ballast is only partially understood, but two important facts are known. First, the behaviour is not primarily attributable to enhanced skin friction, either quasi-steady or unsteady. Second, the greater the volume of air cavities within the gravel, the greater the damping effect.

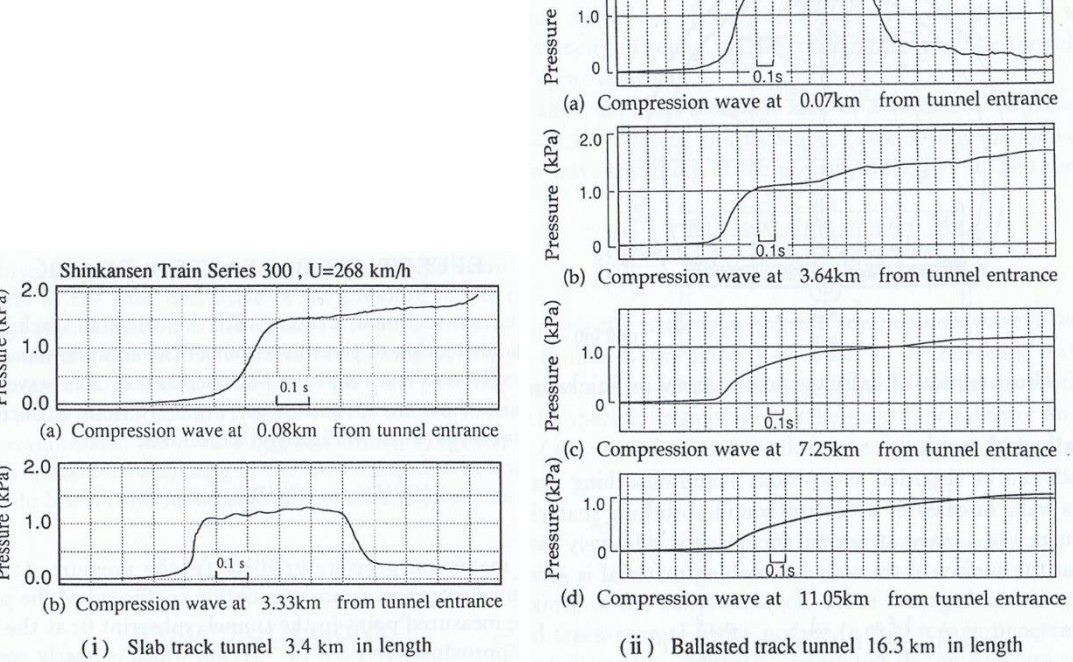

**Figure 8.** Compression waves in long, ballast-track and slab-track tunnels (after Ozawa et al. [16]. N.B.: The strong pressure reduction at 3.33 km in the slab-track tunnel is a reflection from the tunnel exit portal whereas the strong reduction at 0.07 km in the ballast-track tunnel occurred when the train nose passed the sensor).

A partial explanation for this behaviour can be inferred by focussing on what happens at any typical location along the tunnel. As the wavefront passes the location, the pressure above the gravel increases rapidly and begins to force air laterally into the bed, thereby compressing the air trapped in the cavities in the gravel. Resistance to the flow into the ballast layer causes delays in the pressurisation process so the equilibrium of the pressure does not exist until some distance behind the wavefront. The rate of the flow into the ballast depends upon the pressure difference between the air above and within the bed. This difference is negligible at the leading and trailing edges of the ramp. It reaches a maximum where the overlying pressure is approximately half of the value just after the steep nose-entry ramp. Although incomplete, this explanation of one cause of the ballast behaviour has recently been used to propose a method of reproducing ballast-like behaviour in slab-track

tunnels (Liu et al. [17]. That is, research targeted at explaining why a problem does not exist in one type of tunnel has led to a method of minimising the problem in tunnels where it does exist.

As an historical aside, it may be of interest to know that the risk of sonic boom-like disturbances being emitted from railway tunnels was not recognised until it had actually happened. It was first encountered during the commissioning of the first phase of the Shinkansen railway in Japan around half a century ago. At that stage, it was immediately observed that the problem existed only for slab-track tunnels, not for ballast-track tunnels. Of additional interest, the problem with the slab-track tunnels was overcome by building short tunnel extensions with porous walls. Readers might enjoy the challenge of figuring out why, at least so far, it has always been more beneficial to provide these at tunnel entrances than at their exits, where they would act as silencers.

## 7. Delayed Reflections

The reflection process of a pressure wave at an open end of a pipe—connected to a large reservoir, say—is rapid, but it is not instantaneous (Rudinger [18]). Instead, differences between the impedances in the pipe and the reservoir cause interactions that need time to decay. Disturbances propagating from the pipe into the reservoir radiate in a spherical-like manner whereas the reflections along the pipe approximate closely to planar. The time required for the changes in pressure at the outlet to die away is short—typically in the order of the time required for a wave to travel one pipe diameter—so the phenomenon is justifiably neglected in many practical applications. However, this is not always the case, notably when the incident wavefront has significant high-frequency components with wavelengths that are not much longer than the pipe diameter, as is the case in some musical instruments, for instance.

The upper row of Figure 9 depicts the pressure histories at distances of 5, 4, 3, 2 and 1 diameter from the outlet plane of a duct connected to a large reservoir. The rising limbs of the curves show the progress of a steep (near-shock-like) wavefront approaching the reservoir and the falling limbs show its reflection back along the duct. In case (a), the reflection process is assumed to be instantaneous whereas, in case (b), allowance is made for the amplitudes of the induced disturbances in the reservoir to decay. To a close approximation, the curves showing the progress of the reflected wave in case (a) are reversed and inverted images of those showing the incident wavefront. In contrast, the reflected shapes in case (b) are extended in time, strongly so during the later stages of the reflection. As a rough guide, the delays during the first two-thirds of the reflection are less than the time required to travel on the duct diameter, but they increase strongly during the later stages of the reflection.

The lower row of the figure shows the corresponding behaviour for a wavefront that is approximately 1.5 duct diameters long. This is again very steep, albeit less so than the first wavefront. Case (c) shows predictions based on an instantaneous reflection and it is seen that the maximum pressure at a distance of one diameter from the outlet plane is smaller than at greater distances. This is a simple consequence of an overlapping of the later stages of the incident wavefront with the early stages of the reflection. The same overlapping effect necessarily also occurs in case (d) which allows for the delayed response caused by the reservoir. However, the delays allow the pressures close to the outlet to exceed those predicted on the assumption of instantaneous reflection processes.

Tijsseling [19] used an example such as this to disprove a hypothesis put forward by other authors who had postulated much stronger delays and had argued that they could be a strong cause of damping observed in the measurements of water-hammer in pipe flows. Tijsseling correctly reasoned that the timescales of the reflection process were far smaller than those of the discrepancies that the original authors were attempting to explain. In a nutshell, the reflection process does cause delays and damping, but the amplitudes of the delays are very small in comparison with those caused by the other effects discussed above.

For completeness, it is noted that a similar behaviour exists when pressure waves encounter any change in the cross-sectional area; e.g., at a junction of a pipe with a downstream pipe of a larger diameter. However, the differences between the impedances of the two pipes are even smaller than those at a pipe outlet into a reservoir so the induced reflections are smaller. Likewise, the geometrical discontinuity is smaller and so the timescales of the reflection process are smaller. It is almost never useful to take account of this form of damping in the case of liquid-filled pipes.

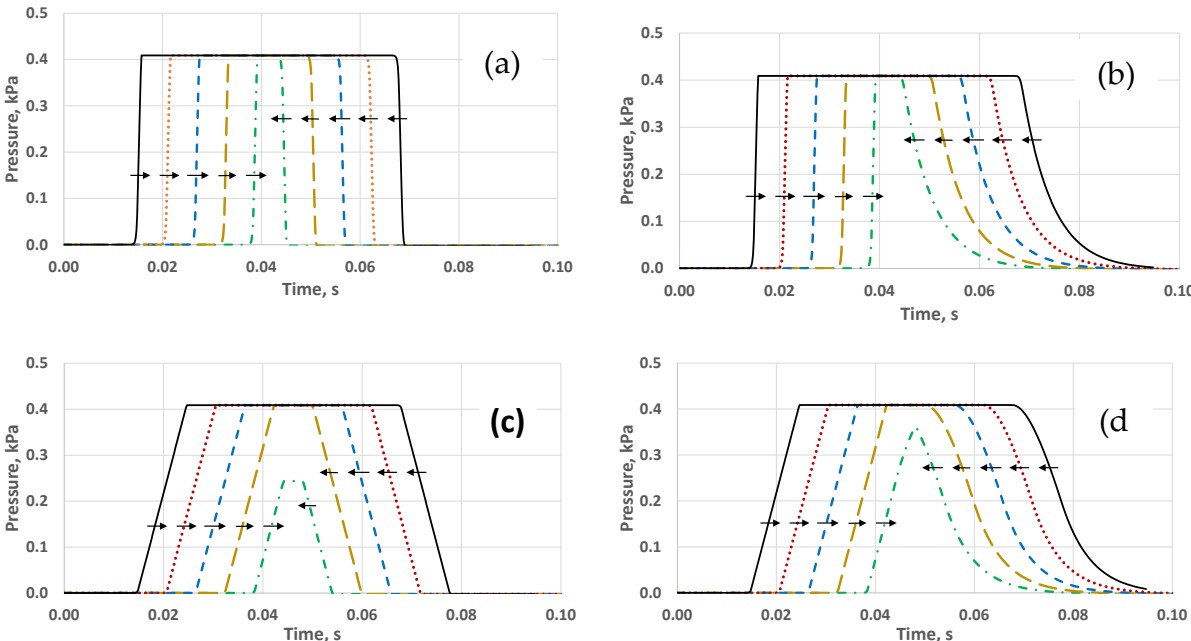

**Figure 9.** Pressures at successive locations as a wavefront approaches and reflects from a reservoir. (**a**) Steep ramp, assuming instantaneous reflections. (**b**) Steep ramp, allowing for true reflection delays. (**c**) More gentle ramp, assuming instantaneous reflections. (**d**) More gentle ramp, allowing for true reflection delays. (The arrows indicate the direction of travel of the wavefronts along the duct as they pass the locations at which the graphs are drawn.

## 8. Experimental Measurements

Attention now turns to consideration of the implications of the above causes of damping for persons attempting to interpret physical measurements. Such persons include, for example, (i) engineers seeking to understand unexpected behaviour, (ii) researchers seeking to isolate particular phenomena and (iii) developers of new techniques for such purposes as leak detection. In all cases, it is necessary (or, at least, helpful) to have a good understanding of the extent to which nominally secondary effects might complicate the task. When damping is seen to be present, it may be necessary to identify its cause so that its implications for other operating regimes can be assessed. Suppose, for instance, that one particular effect is dominant in the measurements, but another one is also exerting some influence. Without a good understanding of both causes, it would be unsafe to assume that the relative importance of the two effects will be the same in other contexts. As a simple example, skin friction is a secondary effect in many laboratory studies of water-hammer, but it can be the dominant cause of the pressure increase in the period after a valve in a long pipeline is closed rapidly. Indeed, for long pipelines, even the correct interpretation of 'rapid' can surprise the unwary.

Against this background, it is informative to revisit some of the figures presented above, attempting as far as practicable to approach them from the standpoint of persons seeing the traces for the first time and without the benefit of theoretical comparisons. First, however, it is useful to discuss briefly the influence of overlapping wavefronts

resulting from, say, the finite times required for valve closure. Figure 10 shows the pressure histories in a classical reservoir-pipe-valve system when the valve is closed rapidly, but not instantaneously, causing a linear reduction in the flow rate to zero in the time required for a sound wave to travel the length of the pipe. To simplify the interpretations, no damping phenomena are simulated; e.g., the flow is treated as inviscid, and no account is taken of local losses or of time delays at the reservoir. The pressure histories are shown at the ends and mid-point of the pipe and also at its quarter points. However, since the pressure at the reservoir never changes, only four curves are seen. Considering each of these in turn:

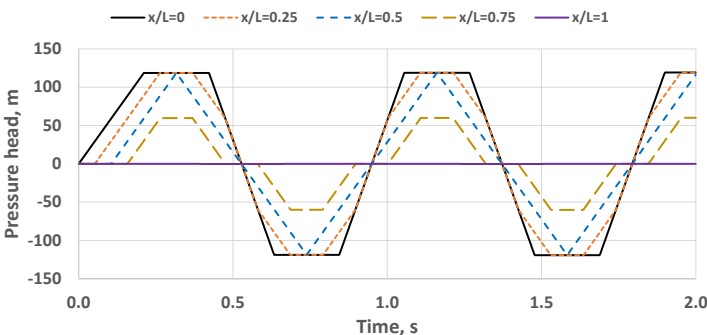

**Figure 10.** Influence of wave superpositions on pressure histories. (Pressures relative to the reservoir pressure.)

- The pressure at the valve rises linearly to a maximum in 0.21 s and then remains constant for a further 0.21 s until the reflection from the reservoir begins to arrive. In the following 0.21 s, it falls linearly to a minimum before remaining constant until the arrival of the next reflection. The amplitudes of the maximum and minimum are equal so the rates of the change in the pressure during the reflections at the valve are double the rate of the change during the closure itself.
- The pressure at $3L/4$ (i.e., at a distance of $L/4$ from the valve) follows a broadly similar pattern except that the changes between the maximum and minimum values occur in three distinct periods. During each reduction, the first period begins when a reflection begins to arrive from the reservoir, and it ends when the subsequent reflection from the valve begins to arrive. During the next period, the pressure change is caused jointly by the second half of the reflection from the reservoir and the first half of its subsequent reflection from the valve. The third period begins at the end of the reflection from the reservoir and continues until the end of the reflection from the valve.
- Next, consider the pressure at the mid-pipe length. At that location, the beginning of any particular reflection coincides with the end of a reflection travelling in the opposite direction. As a consequence, there is no intermediate period of constant pressure, and all rates of change are equal to that in the original closure.
- Finally, consider the pressure history at a distance of $L/4$ from the reservoir. Here, the later parts of each ramp wave travelling towards the reservoir coincide with the earlier parts after they have reflected at the reservoir and, as a consequence, the maximum pressure is smaller than at locations further from the reservoir. Conversely, it is greater than that at locations closer to the reservoir.

A simple message to be taken from this description is that wave superpositions can have a strong influence on pressure histories at any particular location. In this example, each ramp is linear, so the effects are easily seen. In practical applications, however, wavefronts rarely approximate closely to linear and the consequences of superpositions can be less obvious. Another important observation from Figure 10 is that the magnitudes of successive maxima and minima are not affected by the superpositions, and neither are the rates of the change in the pressure of the individual wavefronts. Therefore, although the superpositions can complicate the interpretation of the physical measurements exhibiting

damping, they are not, in themselves, a cause of the damping of individual waves even though, at one location illustrated in Figure 10, they cause a big reduction in the amplitudes.

### 8.1. Re-Assessment of above Figures

Figure 2 is now revisited from the standpoint of an engineer who has only the physical measurements and a description of the reservoir-pipe-valve system. Even though no theoretical curves are given to show the predictions in the absence of damping, it can readily be deduced that strong damping exists. The gradual increase in pressure at the valve during the period of $2L/c$ between the valve closure and the beginning of the first reflection is consistent with the normal expectations of the consequence of a frictional pressure gradient along the pipe in the original steady flow. However, the strongly curved shape in the laminar flow case during the subsequent period of $2L/c$ cannot be attributable to quasi-steady friction. In principle, neither can it be attributed with total certainty to any other particular cause. Nevertheless, since the pipe was encased in concrete, the effect would probably be correctly expected to be caused by unsteady fiction. The measured pressures in the turbulent flow case exhibit the same generic behaviour so the same inferences can be drawn, although the effect is much weaker, so it is obvious until after several reflections have occurred.

A simple comparison of the measurements at the valve and mid-pipe reveals a continual increase in the damping effect as the main wavefront propagates. The most obvious effect is the continual reduction in the amplitudes of the pressure changes. A more important observation, however, is that the individual changes occur successively more slowly. This is clear evidence of a dispersive behaviour. However, it is not easy to assess the rate of the dispersion because the measured values are influenced by the superposition of the reflections such as that described above (Figure 10). This is apparent from the differences between the pressure amplitudes at the valve and at the mid-length of the pipe (n.b., compared with the pressures at $L/4$ in the above example with the inviscid flow). The effect can be seen in both flow cases, although it is less strong in the turbulent flow case than in the laminar flow case.

In summary, quite a lot of valuable information can be deduced from these particular measurements even without the benefit of theoretical predictions with which to make comparisons. However, it is far from common for such clear deductions to be drawn. To illustrate this, consider again Figure 3 and, for this purpose, imagine that the continuous lines had been obtained by physical measurements. In the case of the rapid valve closure, the curve exhibits similar features to those in Figure 2 so it is likely that similar deductions would be made from it, and reasonably so. Nevertheless, the degree of confidence that would be justified in the deductions would be less strong. One reason for this is that measurements are available only at one location. As a consequence, the distortion of successive cycles cannot be proven to increase during propagation along the pipe (even though other possibilities might seem implausible). Likewise, although the existence of superposition effects seems highly probable, the confirmation that would be provided by additional measurements at one or more locations along the pipe is not available.

Notwithstanding the reduced evidence, it is likely that correct inferences would be drawn from measurements showing as much detail as those for the rapid valve-closure case in Figure 3. The same is not true, however, for the slower valve-closure case. For this, it would not even be possible to deduce details of the valve-closure process itself, including, for instance, its start time and, more importantly, its duration. As a consequence, it would not be safe to use the measurements as a basis for predicting, say, what would happen in the event of more rapid closures. In contrast, deductions from measurements made with closure times smaller than $2L/c$ could, with care, be used to estimate likely outcomes for either slower or faster closures. It is somewhat ironic that the case that appears to exhibit greater damping is the one that gives fewer clues about the causes of the damping and, hence, about its wider implications.

Next, consider Figure 4. Once again, all of these pressure histories are theoretical predictions, but, for present purposes, imagine that the physical measurements of the pressure in a reservoir-pipe-valve system yielded the dotted (red) curve labelled QS + US + VE and that it were the only available history. Temporarily disregarding the first $2L/c$; i.e., before the arrival of the first reflection from the reservoir, it could be tempting to attribute the subsequent behaviour to strong unsteady friction. The detailed shape of any particular cycle differs from that displayed in Figure 2, for instance, but this could be imagined to be a consequence of unknown factors such as the characteristics of the valve and the closure sequence. However, consideration of the first $2L/c$ should swiftly dispel this direction of thinking. During this period, the pressure reduces continually and yet, as seen above, if skin friction were the cause of the change, the pressure would increase, not decrease. It would not be plausible to attribute the measured decrease to a negative pre-existing gradient, so it has to be due to some other cause.

This is a strong clue that the pipe itself is responding in a gradual manner to the pressure increase caused by the valve closure. That is, the cross-sectional area is increasing and is continuing to do so even though the pressure is reducing. This implies that the pipe material is behaving in a manner such as visco-elastic or visco-plastic. The direct evidence of this behaviour ceases at $2L/c$, but it would be inferred from the figure that it would have continued for a longer time if the reflection from the reservoir had not interrupted the process. However, there is insufficient evidence to be confident that the strongly dispersive behaviour is attributable to a delayed response of the pipe wall material to imposed pressure changes. In principle, other changes such as, for example, a non-uniform axial distribution of gas bubbles could be responsible. Therefore, anyone needing to identify the true cause of the strong dispersion would need to seek additional information to supplement the measured pressure history.

As a final example, consider again Figure 6, which shows more complex pressure histories than those in other figures. As indicated above, the influence of FSI in this example is far greater than in typical practical situations encountered by either researchers or most practising engineers. Nevertheless, the figure does serve to illustrate the influence that can be exerted by phenomena that are overlooked when assessing measured pressure histories. Recall that the largest sudden pressure change seen in Figure 6b (i.e., L2) exists because the remote end of the pipe was free to move axially. A very different outcome would have resulted if the end had been restrained, but even if it were possible to prevent all axial motion, however small, there would still have been a small reflection because of radial elasticity. FSI will rarely have a sustained influence in most practical applications, but it certainly has an influence when movement is possible, and this will inevitably complicate inferences about other aspects of pressure histories. In practical assessments, it is likely that FSI effects will not be recognised and will either be interpreted as noise or be attributed to other causes. The real point here is that multiple effects coexist in real pressure measurements, and each will contribute to outcomes, especially close to the wavefronts. One consequence is that the interpretation of such measurements is a specialist skill. Another is that researchers need to think broadly when designing experiments targeted at the investigation of specific phenomena. The author wishes all of them much satisfaction in the exercise of their skills.

### 8.2. Unidentified Oscillations

Experimental measurements of conditions following rapid valve closure are commonly used for the assessment of theoretical models of water-hammer phenomena, as in Figure 6 above, for example. In such cases, it is necessary to decide how to represent the valve boundary in the associated theoretical comparisons. One option is to use the measured pressure history directly. Another is to attempt to model the valve itself. Yet another is to idealise the measured history in some way, perhaps using a smoothing technique to reduce the signal noise. Each of these methods has its proponents, but none is strictly rigorous. For instance, the direct use of measured histories should not continue after significant

reflections have begun to arrive from along the pipe. This is not necessarily a significant restriction, however, because, after closure, a valve can be modelled by stipulating zero flow (unless it is itself able to move). A particular concern arises when the measured signal immediately after closure exhibits high-frequency oscillations. It might not be possible to deduce with confidence whether these are attributable to a hydraulic cause induced by the vibration of the valve or to a measurement error in the pressure sensor. Either way, the analyst needs to decide whether and/or how to allow for them.

Suppose that such oscillations exist but are not seen in subsequent reflections. In principle, this could be because (a) physical processes have damped them out or (b) they were due to sensor errors. Now, further suppose that the oscillations are input faithfully to a numerical simulation but are still not seen in the predicted reflections. This could be an indication that a correctly modelled physical phenomenon has damped them out. Alternatively, it could be a consequence of non-physical numerical damping (see the next section). Equally, it could be because they have become obscured by some other phenomenon. The point being made here is that it can be very difficult indeed to infer the validity of a theoretical method when the initial triggering event is subject to uncertainty. This is one reason why the present author considers that descriptions such as 'good agreement' or even 'excellent agreement' can convey misleading messages to unwary readers.

## 9. Numerical Damping

So far, only physical causes of damping have been considered. However, it would be remiss to close the discussion without also commenting on non-physical damping in numerical simulations, which can be important supplementary tools in the interpretation of measured physical behaviour. Almost all numerical simulations are affected to some degree by this problem. For example, it can be a consequence of using numerical grids with regions that are too coarse to propagate higher frequencies correctly. Alternatively, it can be caused by interpolation algorithms or by incorrect estimations of flux across interfaces between adjacent cells. Numerical damping can also be introduced intentionally for special reasons; for instance, by the use of 'artificial viscosity' to suppress unrealistic oscillations close to locations of especially rapid change.

Numerical damping is especially significant when its existence is not recognised. This is a particular issue when simulations are used to provide guidance on the interpretation of experimental measurements. It is all too easy for numerical damping to cause behaviour that enhances apparent agreement between predicted and measured results. This has the potential to cause true physical damping to be interpreted as a consequence of some other effect that a numerical model has been designed to simulate. However, potentially adverse consequences of numerical damping are certainly not limited to the interpretation of measurements. In one especially severe case, an editor over-ruled several reviewers' objections to a paper that presented results from a large parametric study designed to assess the influence of a range of parameters. The paper was published even though all predictions at higher frequencies tended to asymptotic conditions that were demonstratively physically impossible. This example is mentioned in the hope that it will serve as a warning to readers of this present paper. It was especially unfortunate because a key objective of the paper was to enable practising engineers to assess when they needed to take account of the various parameters under study. It is to be hoped that the paper was not widely read.

## 10. Summary and Conclusions

A number of possible causes of the damping of pressure waves in pipelines have been described, with special attention paid to characteristics that researchers and designers might encounter in assessments of measured pressure histories. This can be important when attempting to understand unexpected behaviour, perhaps as a prelude to implementing mitigating countermeasures. It is also important when there is a need to know whether it is safe to use measured results in one pipe system as a basis for predicting conditions to be expected in other systems.

Attention has been paid to damping caused by the response of skin friction on pipe walls to pressure waves causing rapid changes in the axial mean velocity. It has been shown that the time required for vorticity diffusion over a pipe cross-section can cause large differences between the wall stresses and their corresponding values in steady flows. However, the differences reduce rapidly after a wavefront has passed and have little impact on sustained damping. Nevertheless, the effect can have a strong influence on the interpretation of laboratory measurements. Commonly, pipe lengths in laboratories are such that the influence of unsteady skin friction on measured pressure histories is much stronger than in longer pipes typical in engineering practice. This is because the superpositions of elongated wavefronts with their own reflections travelling in the opposite direction can cause continual reductions in the amplitude that would not be seen in most regions of long pipes.

Similar behaviour can arise as a consequence of the visco-elastic nature of many 'plastic' pipes. Sufficiently close to a wavefront, the visual effect can resemble that caused by unsteady friction, although its magnitude is greater. In common with unsteady friction, complications can arise as a consequence of wave superpositions when measurements are made in short pipes. Moreover, the decay times are longer and so the periods of significant overlap are greater. In addition, visco-elastic pipes tend to be much more flexible than metal pipes and this complicates comparisons between the two because of its influence on effective wavespeeds even after the non-linear contributions of the phenomena have decayed.

Another potential cause of damping considered above is bubbly flows. Even low concentrations of bubbles can be influential because they cause wavespeeds to be pressure-dependent and, hence, cause dispersion. This can be especially strong if the gas concentration is not uniform along the pipe. Nevertheless, if the gas is not soluble in the liquid, the process is not inherently dissipative and, although the pressure amplitudes may decrease in some locations, they could increase in others. Therefore, it is debatable whether decreased amplitudes arising from this cause should be regarded as damping per se. Similar comments apply to the case of gas flow in ducts with porous walls. Again, the influence is especially strong when the porosity varies along the duct.

A slightly more complex case arises when pipes have significant freedom to move axially or laterally, as is the case, for instance, with suspended pipe systems. Then, FSI phenomena exist, and their importance will depend strongly on the interactions between internal pressure forces and forces due to structural movements. Once again, it is possible for this to cause decreased pressure amplitudes in some locations, but increased amplitudes in others. In addition, however, the interactions can lead to pressure histories that are much more complex than those in immovable pipes and this inevitably complicates the identification of any truly damping phenomena that might also be present.

For completeness, attention has also been paid briefly to one phenomenon that has been cited as having a significant influence on damping even though it is physically incapable of doing so. This is the reflection process at an open end of a duct; e.g., at the junction of a pipe and a reservoir. Time delays are indeed inevitable as a consequence of the time required for the reservoir to exercise its dominance in the sustainable pressure at the pipe outlet, but the delays are typically shorter than that required for a wavefront to travel one pipe diameter. This is a much shorter delay than those associated with any other of the phenomena considered above.

**Funding:** This research received no external funding.

**Institutional Review Board Statement:** Not applicable.

**Informed Consent Statement:** Not applicable.

**Data Availability Statement:** Not applicable.

**Acknowledgments:** The author wishes to express special thanks to Anton Bergant, Emeritus Head of Applied Research and Computation Department, Litostroj Power, Slovenia and John Vítkovský, Senior Hydrologist, Department of Environment and Science, Queensland Government, Australia, for searching their records to provide experimental and theoretical data illustrating unsteady friction and visco-elasticity.

**Conflicts of Interest:** The author declares no conflict of interest.

## Nomenclature

| | |
|---|---|
| *c* | speed of sound |
| *D* | diameter |
| FSI | fluid–structure interaction |
| *L* | Length |
| *p* | pressure |
| QS | quasi-steady skin friction |
| *t* | time coordinate |
| US | unsteady component of skin friction |
| VE | visco-elastic |
| *x* | axial coordinate |

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
