# Peer review of "On Sources of Damping in Water-Hammer"

_water, doi:10.3390/w15030385_

Round 1
Reviewer 1 Report
Review of:
Title: On sources of damping in water-hammer.
This paper mainly discusses various potential causes of pressure waves attenuation in water-hammer flow, Most of the paper is devoted to presenting causes of damping and to describing their consequences. with special attention being paid to their qualitative influences on measured pressure histories. To avoid unnecessary complications, the discussion centres on two simple geometrical configurations. First, consider the effects of skin friction, viscoelasticity, tube wall properties, bubble flow, porous tube walls, delayed reflections and physical experimental measurements, etc. Finally, the influence of non-physical damping in the numerical simulation is considered for reasons other than physical factors, this paper is completed, and the aspects considered are more comprehensive.
I read this article carefully.
The author grasps a key point for in-depth research, allowing readers to systematically learn some professional knowledge, I suggest that the authors consider a revision of their work along the following suggestions and questions.
Suggestions and Questions:
1-It is recommended to use more easy to understand language in the introduction of the classification to increase the reader's interest and understanding ability.
2-Whether the ‘characterising’ word in line 66 is misspelled, it is recommended to re-check.
3-Whether the discussion of non-physical damping in numerical simulation is relevant to simulation software, it is recommended to consider this issue.
4-This article mentions that only the influence of a single factor is considered, and the influence of several factors is not considered at the same time. Is this comprehensive?
5-It is recommended to add relevant pictures to increase readers' intuitive understanding.
6-What are the advantages and disadvantages of this article, and whether there is a way to improve the defects.
7-Whether the introduction order of this article is arranged according to the primary and secondary effects, and whether these reasons have a primary and secondary relationship, it is recommended to consider this issue.
Reviewer 2 Report
I am so delighted to have the opportunity to review your paper.
The paper is well written and addresses an important issue in water hammer. This will be a very interesting article for the water hammer community. I just suggest a few points but insist that the author can include or discard them.
1. In the introduction section: A more general view to the damping in water hammer can be damping of the water hammer energy (rather that only the pressure head) as defined in [1]. According to this article, energy generated during a transient event at the boundary of a control volume for a given transient duration, is equal to the sum of the kinetic energy and internal (or potential) energy (integrated over the control volume).
2. Regarding the energy dissipation, and its role in damping, can the high frequency waves (that need 2D, 3D modeling) whose energy is dispersed, be another source of damping?
3. Section 2, Skin friction: Regarding unsteady friction (UF), it is beneficial to discuss the impacts of the UF in the frequency response because of the widespread use of FRF for leak and defect detection.
4. The last passage of Sec. 3, (line 286-290). A reference to the following paper is very much informative. “What is wave speed? Tijsseling, A. S. & Vardy, A. E., 2016, BHR Group - 12th International Conference on Pressure Surges. BHR Group Limited, p. 343-360 18 p.”
5. In sec. 3, like in point 3 above, a short sentence to view the linear Viscoelasticity effect in the frequency domain (that VE effect of the pipe walls produces a frequency-dependent wave speed… ) will be helpful. Reference to this paper can be made: “Frequency response of water hammer with fluid-structure interaction in a viscoelastic pipe HK Aliabadi, A Ahmadi, A Keramat, Mechanical Systems and Signal Processing, 2020”
Once again, the paper is great, and it is up to the author’s choice to include the raised comments.
Reviewer 3 Report
This paper aims to give an overview of the damping sources in water hammer for pipe flows by discussing the main effects, including skin friction, visco-elasticity, bubbly flows, and porous pipe linings. The discussions are quite comprehensive and to certain depth. This paper would be very beneficial for the initiate who tend to design pipelines or evaluate pipe flows. The writing is very clear and in logical manner. This paper is recommended to be published in Water after two parts have been revised:
Line:114: There is no Fig.1b. So is it a typo?
Figure 6: Please add unit to (a) and labels for both axes to (b) and (c).
Round 2
Reviewer 1 Report
accept